# Biomolecules Involved in Both Metastasis and Placenta Accreta Spectrum—Does the Common Pathophysiological Pathway Exist?

**DOI:** 10.3390/cancers15092618

**Published:** 2023-05-05

**Authors:** Anna K. Rekowska, Karolina Obuchowska, Magdalena Bartosik, Żaneta Kimber-Trojnar, Magdalena Słodzińska, Magdalena Wierzchowska-Opoka, Bożena Leszczyńska-Gorzelak

**Affiliations:** Chair and Department of Obstetrics and Perinatology, Medical University of Lublin, 20-090 Lublin, Poland; arekowska@icloud.com (A.K.R.); magdalenaklaudia1@gmail.com (M.B.); magdalenaszepietowska@wp.pl (M.S.); magdalena.wierzchowska-opoka@umlub.pl (M.W.-O.)

**Keywords:** biomolecules, CDH1, epithelial–mesenchymal transition, LAMC2, metastasis, PIGF, placenta accreta spectrum, placenta percreta, VEGF, ZEB proteins

## Abstract

**Simple Summary:**

Both invasive placentation and cancer metastasis involve the intricate migration and invasion of cells into the surrounding tissues, while also creating new blood vessels to support their growth. The complex interplay of molecular signals that regulate cell behavior, including growth factors, cytokines, and extracellular matrix components, plays a crucial role in both processes. Despite their differing physiological contexts and evolutionary origins, invasive placentation and cancer metastasis share many molecular and cellular mechanisms that allow them to invade and proliferate in the surrounding tissues. In both pathological conditions, cells need to overcome the local mechanisms of immunological defense, activate invasion, and induce angiogenesis. This article focuses on describing the biomolecules that link these pathological processes.

**Abstract:**

The process of epithelial-to-mesenchymal transition (EMT) is crucial in the implantation of the blastocyst and subsequent placental development. The trophoblast, consisting of villous and extravillous zones, plays different roles in these processes. Pathological states, such as placenta accreta spectrum (PAS), can arise due to dysfunction of the trophoblast or defective decidualization, leading to maternal and fetal morbidity and mortality. Studies have drawn parallels between placentation and carcinogenesis, with both processes involving EMT and the establishment of a microenvironment that facilitates invasion and infiltration. This article presents a review of molecular biomarkers involved in both the microenvironment of tumors and placental cells, including placental growth factor (PlGF), vascular endothelial growth factor (VEGF), E-cadherin (CDH1), laminin γ2 (LAMC2), the zinc finger E-box-binding homeobox (ZEB) proteins, αVβ3 integrin, transforming growth factor β (TGF-β), β-catenin, cofilin-1 (CFL-1), and interleukin-35 (IL-35). Understanding the similarities and differences in these processes may provide insights into the development of therapeutic options for both PAS and metastatic cancer.

## 1. Introduction

The implantation of the blastocyst plays a crucial role in gestation and fetal development, and the process of epithelial-to-mesenchymal transition (EMT) is central to this. The trophoblast, which is composed of two zones—villous and extravillous—serves different functions. The villous trophoblast acts as a barrier between maternal and fetal circulation, while the extravillous trophoblast (EVT) invades the trophoblast cell column to reach the connective tissue of the uterus.

Transcription factors present in the cellular microenvironment repress genes responsible for the adhesion of epithelial cells, resulting in the loss of polarity in epithelial cells, remodeling of the cytoskeleton, and replacement of cytokeratin by vimentin [1,2]. This leads to the transformation of adherent epithelial-like cells to a migratory mesenchymal phenotype, acquiring infiltrating properties [3].

The subsequent processes of placental development and decidualization require attachment and invasion of the blastocyst to the endometrium [4,5]. These processes are carefully regulated by the decidual microenvironmental compounds that facilitate invasion, angiogenesis, and immune tolerance of the trophoblast cells [2].

In pathological states, EVT can also contribute to the occurrence of placenta accreta spectrum (PAS).

The placenta is a complex, temporary organ composed of two parts: the maternal decidua and the fetal trophoblast [1,2]. PAS is a condition in which the placenta attaches and invades the myometrium, instead of the normal decidualized endometrium, beyond the physiological level of Nitabuch’s layer [2,4,6,7].

There are three types of PAS disorders: placenta accreta, placenta increta, and placenta percreta, which are classified according to the depth of the placental invasion. Placenta accreta is the most common variant, where the placenta only attaches to the myometrium without invading it. In placenta increta, the placenta reaches and invades the myometrium, while in placenta percreta, the invasion reaches the serosal or deeper layers [4].

Although the exact pathophysiology of PAS remains unknown, it is believed to be caused by dysfunction of the trophoblast, imbalanced expression of maternal and placental proteins, or defective decidualization [2,4,6]. Two independent risk factors for PAS are placenta previa and previous cesarean section deliveries [4,6,8]. Over the past five decades, PAS has emerged as one of the most significant obstetric complications, leading to peripartum hemorrhage, organ failure, and disseminated intravascular coagulation (DIC). These result in increased maternal and fetal morbidity and mortality [4,7].

Due to the unique characteristics of the placenta—a semi-allogeneic organ that invades and remodels surrounding tissues and is tolerated by the host immune system—some studies have examined the similarities between placentation and carcinogenesis. Cancer is currently one of the leading causes of morbidity and mortality worldwide, responsible for nearly 10 million deaths annually [9].

Metastatic cancer is defined as the presence of tumorcells that have spread to the surrounding or distant tissues of the body, and as such disseminated disease typically carries a poor prognosis for patients. Metastases are estimated to be responsible for up to 90% of cancer-related deaths [10]. The complexity of the metastatic process and its underlying mechanisms need further investigations to find successful therapeutic options for metastatic disease. However, numerous studies have investigated the conditions of tumor primary invasion and growth, initiation of metastases, cell dissemination, and metastasis formation. Among many mechanisms involved in the metastatic process, there are EMT and the establishment of a microenvironment that facilitates this process [10,11,12].

Given the striking parallels between the pathogenesis of PAS and metastasis, and the limited understanding of both conditions, we conducted a study to explore the molecular biomarkers that are involved in the microenvironment of both tumors and placental cells. In this article, we present our review of the following biomolecules: placental growth factor (PlGF), vascular endothelial growth factor (VEGF), E-cadherin (CDH1), laminin γ2 (LAMC2), the zinc finger E-box-binding homeobox (ZEB) proteins, αVβ3 integrin, transforming growth factor β (TGF-β), β-catenin, cofilin-1 (CFL-1), and interleukin-35 (IL-35).

## 2. Biomolecules

### 2.1. PIGF

PlGF is a member of the VEGF family [13], as we described below in the VEGF paragraph. However, its nature is of great importance in the aim of this article, and we have to focus more attentively on its function. We can distinguish four isoforms made by alternative splicing [14]. In normal conditions, it is produced mainly by the growing placenta, inducing the development of trophoblasts and the maturation of emerging blood vessels. This process is caused by affecting endothelial cells directly, but also indirectly influencing other types of cells [15].

The binding of PlGF to cells is possible through cooperation with VEGFR-1/soluble fms-like tyrosine kinase-1(sFlt-1), neurolipins (which are coreceptors of tyrosine kinase receptors), and heparins [14,16]. PlGF can also stimulate VEGFR-2, but this is carried out indirectly. In pathological situations, PlGF supplants VEGF-A from binding with VEGFR-1 and replaces it with what VEGF-A binds to in VEGFR-2. This action plays a role in inflammation, cancer, and ischemia because it intensifies the properties of VEGF-A for mobilizing and proliferating cells and increasing vessel permeability [13,14,15]. Many cells react to the impact of PlGF. In a healthy state, PlGF is mostly inactivated by sFlt-1, which neutralizes its activity [14].

As speculated, PlGF might induce the growth of expression of other molecules, such as proangiogenic factors (platelet-derived growth factor subunit B(PDGFB), fibroblast growth factor-2 (FGF2), VEGF), matrix metalloproteinases, and those taking part in inflammatory development [15]. These properties can be enhanced in pathological states associated with an increase in the concentration of the active form of PlGF, which may be up-regulated by the appearance of HIF in oxygen deficiency [17]. The interesting phenomenon is described in a study by Patel et al., which shows the possibility of activating HIF-1α from monocytes and epithelial cells mediated by PlGF in hypoxia [18]. The above circumstances occur in high levels of PlGF concentration in cancer tissues, which contribute to their proliferation, progression, local invasion, lymph, and distant metastases through blood flow, poor response to treatment, and implicate poor survival [14]. This occurs in various types of cancers, such as melanoma, small-cell lung cancer, thyroid carcinoma, serous ovarian cancer (OC), and oral squamous cell carcinoma [14,19,20,21,22,23,24,25].

PlGF has been extensively studied in the context of pathological pregnancies, where it has been compared to normal pregnancies. Research shows that the serum levels of PlGF are significantly higher in PAS and PAS subgroups compared to the Normal Placenta group. The PlGF cut-off point derived from maternal serum levels can be used to predict PAS validity and, if used as a screening test in an earlier pregnancy, could lead to more effective management of these pregnancies [26].

Additionally, PlGF has been found to play a role in preeclamptic pregnancies, where incorrect blood vessel development in the placenta is thought to be a contributing factor.

In pathological pregnancies, there is an increase in both placental and circulating sFlt-1, which results in decreased levels of free VEGF/PlGF [27]. Research by Zhang et al. suggests that measuring sFlt-1 and PlGF levels, as well as their ratio, in the blood of pregnant women may be useful in differentiating patients with PAS from those with placenta previa. The study found that PAS patients had higher levels of PlGF and a lower sFlt-1/PlGF ratio compared to the placenta previa group [28].

### 2.2. VEGF

VEGF is a protein belonging to the VEGF family, which is a large group of molecules consisting of VEGF-A, VEGF-B, VEGF-C, VEGF-D, VEGF-E, and placental growth factor (PlGF) [1]. VEGF is a subfamily of growth factors (GF) [2]. VEGF is a multitasking protein that mediates both vasculogenesis (the process of blood vessel formation from endothelial progenitor cells in embryos and adults during tumor growth) and angiogenesis (the formation of blood vessels from pre-existing vessels formed during vasculogenesis, consisting of sprouting and splitting) [13,29,30].

Types of VEGF family members [13,19] include:VEGF-A (also called VEGF) is the best-known and most important factor, mainly involved in vasculogenesis and angiogenesis. The main targets of VEGF-A are endothelial cells, in which it stimulates migration and mitosis, inhibits apoptosis, and dilates vessels with NO. There are many specific isoforms of VEGF-A, with various features, formed during alternative splicing. VEGF-A is secreted by kidney mesangial cells, macrophages, cells of the retina, osteoblasts, keratinocytes, platelets, and many others. VEGF-A is bound to extracellular matrix elements, and proteolytic enzymes (metalloproteinases and plasmin) can release free, diffusible forms of it, which are active in the extracellular environment.VEGF-B is primarily involved in embryonic tissues in the development of the cardiovascular system. In adult individuals, it takes part in myocardial remodeling.VEGF-C initiates the development of lymphatic tissue during lymphangiogenesis but is not a strong angiogenic factor.VEGF-D, similar to VEGF-C, controls lymphangiogenesis, but in the lungs.VEGF-E (viral) is not found in humans.

Different types of VEGF require various types of VEGF receptors. All of them are tyrosine kinase receptors, consisting of three parts: a transmembrane domain composed of the cell membrane, an extracellular domain for binding VEGF from ECM, and an internal domain involved in the phosphorylation of tyrosine. Types of VEGF receptors include [13,19]:VEGFR-1 (Flt-1) is responsible for binding VEGF-A, VEGF-B, and PlGF. VEGFR-1 has also been produced in the ECM as a soluble isoform (sVEGFR-1/sFlt-1) and, similarly to the transmembrane type, can bind the same factors. Surprisingly, sFlt-1 can cooperate with VEGFR-2 to decrease its activity. Consequently, sVEGFR-1 employs anti-angiogenic, anti-edema, and anti-inflammatory activities, and its dysregulation has been connected with other pathological processes. The pathogenesis of preeclampsia, which usually occurs in the last trimester of pregnancy and is linked to sVEGFR-1 production because of the placenta, and subsequent neutralization of VEGF-A and PIGF signaling. A poor quantity of sVEGFR-1 to VEGF-A has been tied to excessive tumor malignancy/invasiveness and inferior patient survival. Additionally, sVEGFR-1 may play a proangiogenic and protumoral role as well, through the activation of β1 integrin, which stimulates endothelial cell adhesion and chemotaxis [31].VEGFR-2 (KDR/Flk-1) binds VEGF-A and, on special occasions, VEGF-E, C, and D. Its main function is to initiate vasculogenesis and can be found not only on epithelial cells but also on hemangioblasts.VEGFR-3 (Flt-4) is a receptor for VEGF-C and VEGF-D and is a mediator in the lymphangiogenesis process.

Tumor angiogenesis plays a significant role in cancer progression and metastasis [13,19,30]. The development of a tumor mass requires oxygen and nutrient delivery, which necessitates the growth of new blood vessels. This problem is partly solved by increasing the expression of VEGF/VEGFR produced by cancer cells [6]. Therefore, as VEGFR-1 has a crucial role in tumor-associated angiogenesis, it is also involved in physiological angiogenesis [19]. Pathological lymphangiogenesis caused by the connection between VEGF-C/VEGF-D and their receptor (VEGFR-3) is also involved in tumor progression. New networks of lymphatic vessels create opportunities for metastatic cells to spread to distant tissues and lymph nodes. The expression VEGF has many regulatory factors, one of which is the hypoxia-inducible factor (HIF), which is triggered by a hypoxic environment [32]. Besides VEGF, HIF can also activate more proangiogenic agents such as angioproteins, TGF-β, TNF-α, and basic FGF [13]. The last-mentioned one can induce VEGF synthesis, and together they have synergy in action. Decreased pH, low levels of glucose, hypertension, and inflammation can also intensify the production of VEGF and its receptors. In metastatic processes, other molecules are involved, mainly those that disturb the stability of cell junctions, such as cadherins and matrix metalloproteinases, which are induced by VEGF. These signaling pathways are commonly documented in various types of cancers originating from different organs and tissues, e.g., lungs, liver, skin, brain, kidneys, pancreas, bones, gastrointestinal tract, bone marrow, breast, OC, and prostate [19,33,34,35]. According to a study by Liang et al., the proliferation of cells from various breast cancer cell lines and inhibition of apoptosis by increasing Bcl-2 levels are connected with VEGF secretion. A higher concentration of VEGF, probably bound to sex steroid hormones, is associated with a poor response to hormonal therapy in treatment and worse survival [35,36].

Implantation of the embryo during pregnancy leads to the formation of a new organ, the placenta. Trophoblast villi invade the top layer of the uterus, called the decidua. Proper placental development also involves the development of new blood vessels. Extravillous cytotrophoblast cells proliferate and interact with spiral arterioles in the uterine wall, subsequently remodeling them and incorporating them into the epithelial cells. This creates a connection between the maternal and fetal blood supplies necessary for exchanging substrates for fetal growth and metabolic products produced by it [37].

Angiogenesis in the placenta and tumors is based on similar pathways, including VEGF, PlGF, and their receptors, and their levels are regulated by various hormonal and nonhormonal agents. However, it is worth mentioning that the level of PlGF in normal placental development is significantly higher compared to VEGF levels [16]. Disorders in the correlation between levels of pro-angiogenic factors, receptors, and their soluble forms may be one of the reasons for PAS. Tseng et al. studied the differential expression of vascular endothelial growth factor, placenta growth factor, and their receptors in placentae from pregnancies complicated by PAS. The results of their work showed no difference in the content of PlGF and sFlt-1 in samples taken from patients with disturbed and undisturbed placentation. In the second conclusion, they suggested that women with PAS have a lower quantity of sVEGFR-2 and higher expression of VEGF than the control group [16].

Wang et al. also dealt with similar issues exploring the association of VEGF and sFlt-1 and their use for the diagnosis of pernicious placenta previa (PPP) and this condition complicated by placenta accreta/increta [38]. The authors indicated that VEGF was negatively correlated with sFlt-1 in the serum of PPP patients. Moreover, numerous past experiments showed that the decrease of VEGF and its receptors and the increase of sFlt-1 in serum could specifically indicate the presence of placenta accreta. According to the article, similar results could be found. The authors discovered that VEGF concentration was the lowest in PPP correlated with placenta increta and continuously increased in PPP combined with placenta accreta, PPP alone, and healthy controls, and there was a remarkable variance between the groups. Therefore, sFlt-1 had the highest concentration in the PPP combined with the placenta increta group, followed by PPP combined with placenta accreta and the PPP group, which had the lowest concentration in the control group [38].

Disorders in proangiogenic factors can lead to not only invasive growth of the placenta in PAS but also other conditions with faulty placenta development. As we know, increased VEGFR-1 is associated with a higher sFlt1 concentration, which plays a negative role in angiogenesis by binding VEGF-A and PlGF [13,16,39]. Preeclampsia is a disorder that mainly occurs in the third trimester of pregnancy and is a life-threatening medical condition for both mother and fetus [19]. The main symptom is higher than normal blood pressure, which indicates incorrect functioning of other organs and may lead to proteinuria, liver dysfunction, visual disturbances, and other complications [40]. Research shows an increased sFlt-1 level in patients with preeclampsia [13,19,39,40,41].

### 2.3. CDH1

CDH1 is a member of the cadherin family—cell adhesion transmembrane glycoproteins dependent on calcium, important in forming adherens junctions between cells [42]. The cadherin superfamily includes cadherins, protocadherins, desmogleins, desmocollins, and more. CDH1 belongs to the aforementioned family and is located on epithelial cells, mainly deployed in zonula adherens junctions [42,43].The molecules of CDH1 consist of three parts: extracellular, transmembrane, and cytoplasmic [43]. The extracellular domain, changing from cis-dimers to trans-dimers, plays an important role in cell-to-cell adhesion. However, the intracellular part plays an important role in binding p120-catenin, β-catenin, and alpha-catenin [44]. This complex formation stabilizes epithelial tissues, regulates intercellular exchange, and links to the actin cytoskeleton through its interaction with the catenin complex [42,44].

CDH1 is an anti-oncogene. Its main role is to regulate proliferation and cell division during the cell cycle, which is important for the appropriate formation of new tissues. Disruption of this process can lead to EMT, which can in turn lead to carcinomatosis [45,46]. EMT can be induced by many oncogenic pathways, such as transforming growth factor-β (TGF-β), MAPK, Ras pathway, ERK, PI3K/Akt signaling, and cyclin kinase inhibitor p27-mediated signaling [47,48,49]. Reduction or lack of CDH1 expression has been observed in tumors and metastases of many organs, including the gastrointestinal tract, urinary tract, uterus, ovary, thyroid gland, prostate, and skin [43,45,50,51]. However, loss of CDH1 is not always necessary for EMT in human breast cancer, as reported by Hollestelle et al. [45,52]. Recently, the potential role of CDH1 in metastasis as a prognostic indicator has been investigated, as loss or reduction of its expression correlates with increased aggressiveness of cancers [43].

Some tumors have been observed to regulate the expression of CDH1 during the metastatic process, raising questions about the role of the tumor microenvironment in promoting or inhibiting tumor cell migration and invasion [43].

In oral squamous cell carcinoma, research by Lorenzo-Pouso et al. shows that altered expression of CHD1 in cells can serve as a prognostic biomarker for survival [51]. The main function of malignant tumors is to transform the environment and progress through cell proliferation. Malignant tumors are characterized by a disorganized architecture that hinders the efficient removal of waste products, creating a toxic environment that promotes the uncontrolled proliferation of transformed cells. Poorly adherent cells, which are transformed and have decreased expression of CDH1, can invade surrounding tissues and metastasize.

In this vicious circle, due to hypoxia, peptide growth factors such as TNF-α and EGF are released, which also secondarily reduces CDH1 expression [43].

Despite the commonly held belief that loss of CDH1 expression is a key factor in metastasis, this oversimplification ignores the fact that many metastases still contain high levels of CDH1. Furthermore, it is possible for CDH1-expressing epithelial cells to become invasive without undergoing the full EMT process in tumors [45].

It is worth mentioning that CDH1 can promote the invasiveness and progression of tumors in advanced stages of cancer [47]. In some situations, a reverse process called MET occurs, which involves an increase in CDH1 levels and is performed in cancer metastasis [48].

EMT is also involved in placentation, a process crucial in the development of pregnancy that ensures maternal-fetal circulation. By acquiring mesenchymal cell characteristics, epithelial cells of the trophoblast can invade and migrate into the maternal decidua [53].

The continuity of CDH1 in the basement membrane of syncytiotrophoblast plays a crucial role, which is supported by studies on syncytin 1. This cell-cell fusion protein promotes the proliferation of cytotrophoblasts by regulating the cell cycle, while also mediating the fusion of cytotrophoblasts to the syncytium [54].

Several pregnancy pathologies are thought to be linked to anomalous migration and inadequate trophoblast invasion. Shallow invasion, for instance, is a hallmark of gestational hypertension and preeclampsia (PE). The disruption of DCH1 expression continuity in the basal membrane of the syncytiotrophoblast is associated with elevated vascular resistance, the appearance of early diastolic notches in the uterine artery flow waveform, proteinuria in preeclampsia patients, and lower Apgar scores in newborns [53].

Preeclampsia is also regulated by ribosomal protein L39 (RPL39), a ribosomal protein that belongs to the S39E family of ribosomal proteins. This molecule is involved in metastasis, stem cell renewal, and chemoresistance by decreasing the expression of CDH1 and weakening adherens junctions. In preeclampsia, however, the opposite is true; RPL39 is down-regulated and CDH1 is up-regulated, which inhibits trophoblast invasiveness [55].

There is an interesting connection between CDH1 and chronic venous disease in pregnant women. Ortega et al. observed low expression of CDH1 in their research, which may be related to complications such as placenta accreta, placenta percreta, preeclampsia, or gestational trophoblastic disease in these patients [56].

CDH1 is involved in abnormal placentation in pregnancies [46]. Several research papers support the relationship between the loss of CDH1 and placenta accreta or placenta percreta. Incebiyik et al. revealed a significant difference in CDH1 expression between the control group of pregnant women with positive CDH1 staining and patients with placenta percreta, with the latter showing negative staining for CDH1 [57]. El-Hussieny et al. also found a significant decrease in CDH1 immunoexpression in trophoblasts of placenta accreta when compared with control cases [58].

Timofeeva et al. attempted to develop diagnostic methods for the abnormally invasive placenta (AIP), which includes placenta accreta, percreta, and increta [49]. As we know, trophoblasts require EMT to invade the decidua, which is controlled by multiple signaling pathways, as mentioned earlier. The activation of these pathways leads to changes in CDH1 expression, reducing its level and altering cell-to-cell adhesion junctions, ultimately allowing cells to invade and migrate. During the reduction of CDH1 quantity, proteolysis of CDH1 by A disintegrin and metalloproteinase domain-containing protein 10 (ADAM10) and presenilin-1 (PSEN1) lead to the formation of soluble forms of CDH1 that circulate in the blood plasma [46,49]. This phenomenon could be used to differentiate a normal invasive placenta from an abnormal invasive placenta, as proteolysis is aberrant in the latter. However, the results of this research are somewhat divergent.

The concentration and proportion of soluble CDH1 fragments in the blood plasma of pregnant women with any form of acute intermittent porphyria (AIP) were found to be comparable to those in the blood plasma of healthy pregnant women [49].

### 2.4. LAMC2

Laminins are a family of extracellular glycoproteins that consist of 3 polypeptides—chains α, β and γ connected by disulfide bonds. They are one of the main compounds of cell basement membranes and take part in cell attachment, migration, signaling, and metastasis [59]. LAMC2 subunit is a compound of Laminin-5 molecule, which is a subepithelial basement membrane isoform [60].

LAMC2 overexpression was also found in cancer cells and therefore the correlation between tumorigenesis and tumor spread and the protein levels was investigated. It was reported that LAMC2 expression is up-regulated in pancreatic ductal adenocarcinoma (PDAC), non-small-cell lung cancer (NSCLC), penile squamous cell carcinoma (PSCC), OC, anaplastic thyroid carcinoma, oral tongue squamous cell carcinoma, cholangiocarcinoma, hepatocellular carcinoma and esophageal squamous cell carcinoma [59,61,62]. Moreover, LAMC2 overexpression was associated with lymphnode metastasis and tumor-node-metastasis stages [59]. LAMC2 was also identified as a metastasis organ site-specific biomarker in bladder cancer. In pancreatic cancer, they enhance cell migration and invasion [63]. Additionally, LAMC2 promotes metastasis and tumorigenesis in pancreatic ductal adenocarcinoma was induced by activation of EGFR/ERK1/2/AKT/mTOR signaling pathway and repression of EMT [64,65]. Mentioned reports prove the role of LAMC2 proteins in cancer cell migration, cellular phenotype maintenance, adhesion, migration, growth, and differentiation [59]. Delle Cave et al. identified a highly metastatic subpopulation of tumor-initiating cells in PDAC with up-regulated LAMC2. The authors observed a correlation between *CD44* (the stemness-associated gene), and LAMC2 up-regulation in spheres. Moreover, the study explained LAMC2-EGFP+ increased metastatic character, by showing that those cells undergo EMT and show significant migratory potential. Additionally, cytokines in the tumor environment, such as TGF-β1 could induce LAMC2 expression. It was discovered that LAMC2 knockdown diminished migration, invasion, and tumorigenicity [66]. In PSCC, LAMC2 indicates tumor advancement and the occurrence of lymph node metastasis. LAMC2 could be a valuable prognostic marker for early-stage PSCC [67].

Wang et al. found significant overexpression of LAMC2 in placentas of PAS compared to normal placentas [68]. The regulating role of AKT1 in placentation and the negative correlation between AKT1 and LAMC expression have already been described [69,70]. However, the study revealed that LAMC2 was responsible for the moderation of trophoblast migration, proliferation, and inhibition of apoptosis via the PI3K/Akt/MMP2/9 pathway. By that, LAMC2 promotes trophoblast over-invasion, and the occurrence of PAS [68].

### 2.5. ZEB Proteins

In recent years, various studies have analyzed the impact of the zinc finger E-box-binding homeobox 1 (ZEB1) and zinc finger E-box-binding homeobox 2 (ZEB2) homologous transcription factors on carcinogenesis and cancer progression. ZEB1 and ZEB2 proteins are members of the human ZEB family, consisting of two clusters of zincfinger and a centrally located POU-like homeodomain [71,72]. They act as transcription factors and by binding to the CDH1 gene region—CDH1/PKP2 on chromosome 16, repress the adhesion epithelial genes, such as CDH1 expression, in tumor cells [73]. ZEB1 and ZEB2 are among the main transcription factors inducing EMT [71,73,74,75]. Cells undergoing EMT can remodel the basement membrane and, as a result, invade surrounding tissues [73].

ZEB molecules play a pivotal role in enabling the survival of tumor cells, boosting proliferation, invasion, and dissemination [76]. Due to the invasive and promigratory characteristics of EMT, and its clear connection with ZEB proteins, it has become a target to distinguish the direct correlation between metastatic capacity and ZEB expression. ZEB1 induces metastasis in pancreatic, colorectal cancer, breast cancer, and oral cavity squamous cell carcinoma [77,78,79]. Multiple cancer types, including OC, breast, hepatocellular, colorectal, and gastric cancer with overexpression of ZEB2 revealed a correlation with tumor metastasis, progression of the disease, and poor prognosis [72]. In non-small-cell lung cancer, ZEB1 can suppress CDH1, thus promoting metastasis and the activation of the epidermal growth factor receptor [79]. A study on colorectal cancer (CC) cells revealed that ectopic expression of ZEB2 was responsible for increasing proliferation and metastatic potential in vivo and in vitro in HCT166 cell lines. It also promoted angiogenesis in the xenotransplantation models [80]. Song et al. showed the up-regulation of ZEB2 in OC and its involvement in cancer-associated angiogenesis, which induces tumor invasion and metastasis [80]. Interestingly, in melanoma, ZEB2 and Wingless/Integrated (Wnt) are responsible for the induction of microphthalmia-associated transcription factor (MIFT) expression, which is the main regulator of melanocyte homeostasis. Thus, the proliferative status of neoplastic cells and differentiation. However, ZEB1, together with Notch and TGF-β, diminishes MIFT expression, and cells acquire invasiveness and stem-cell-like character. Cells of both primary melanomas and metastases express MITFlow as well as MITFhigh [12]. Moreover, it has been reported that the transformation from ZEB2 to ZEB1 stimulates the development of metastatic melanoma [72]. An important role in ZEB-induced EMT regulation is attributed to microRNA (MiR). ZEB1 suppressed by miR-200a presents increased CDH1 levels, thus inhibiting EMT. Moreover, MiR 1271 by ZEB1 binding can reduce cell proliferative potential and viability. However, miR-30a, miR-3653, miR-138-5p, and miR-377 have been described as neoplasm suppressors using ZEB2 inhibition. However, it was reported that in bladder cancer, circular RNA ZFR (circZFR) inhibits miR-377 and therefore promotes tumor spread. Circular RNA has_circRNA_406752(circPCNXL2) and long non-coding RNA (lncRNA) SNHG5 enhance the expression of ZEB2 and prompt metastasis as well [76].

Post-translational modifications (PTMs) of proteins can also regulate cancer cell proliferation. In colorectal cancer, ZEB2 undergoing glycogen synthase kinase-3 beta (GSK3β) phosphorylation shows migratory potential, while ectopic expression of ubiquitin-specific protease 14 (USP14) is linked to liver and lymph node metastasis. Another agent promoting EMT is ZEB1 deubiquitinated by ubiquitin-specific peptidase 18 (USP18) in esophageal squamous cell carcinoma, while overall SUMOylation is associated with cancer progression [81].

Data published by Ilsey et al. indicates the role of EMT as a mechanism inducing cytotrophoblast differentiation into invasive EVT [82]. Meanwhile, Da Silva et al. proved that ZEB2 has a key role in EMT induction and promotion of cell invasiveness in trophoblast differentiation. Their research also shows that third-trimester EVT from the over-invasive abnormally invasive placenta (AIP) pathologies has a more mesenchymal EMT type compared to physiological EVT [75]. AIP is clinically characterized as placental overexpression [82]. ZEB1 promotes placental implantation and is overexpressed in placenta accreta. Thus, Li et al. hypothesized that ZEB1 could be one of the factors inducing PAS. The authors also hinted at the possible role of the Akt signaling pathway in ZEB1 expression regulation; however, the mechanism is yet to be confirmed [83,84].

### 2.6. αVβ3 Integrin

Integrins are a family of transmembrane cell receptors responsible for cell adhesion, interactions, and signaling between cells and the extracellular matrix (ECM) [85,86,87]. The αvβ3 integrin, also known as the vitronectin receptor, consists of two subunits, αv and β3, and can bind various ECM molecules, including vitronectin, fibrinogen, fibronectin, and proteolyzed forms of laminin and collagen. The αvβ3 integrin is involved in angiogenesis, neovascularization, cell invasion, proliferation, and metastasis [6,86]. The αv and β3 subunits are present in syncytial microvillous membranes and are necessary for binding numerous ECM molecules, including vitronectin, fibronectin, and osteopontin, during implantation.

Weitzner et al. investigated the possible role of αvβ3 integrin in the development of placenta percreta. The authors aimed to determine whether the pro-invasive and promigratory features of αvβ3 integrin could also contribute to the phenomenon of pathologically invasive placentation. Study results revealed significant integrin overexpression in placenta percreta EVT [6].

In general, overexpression of integrin αvβ3 can be detected in rapidly growing endothelial cells and tumor cells and is specifically linked to tumor growth and spread in various cancers, including melanoma, prostate cancer, breast cancer, OC, lung adenocarcinoma, pancreatic cancer, and osteosarcoma [85,87,88,89]. Patients with aberrant expression of αvβ3 are at greater risk of metastatic disease and shorter survival time [90]. Wang et al. reported that the interaction between αvβ3 integrin and sialoprotein (BSP) induces bone metastasis in breast cancer [86,90]. Moreover, Kariya et al. reported that αvβ3 induces partial EMT as another mechanism responsible for metastasis development [89]. Even though tumors with high expression of αvβ3 are more likely to metastasize, inhibition of αv and β3 integrins could suppress cancer spread and become a promising target for cancer therapies [90,91].

### 2.7. TGF-β

TGF-β is a family of pleiotropic cytokines consisting of three TGF-β isoforms—TGF-β1, TGF-β2, and TGF-β3—activin, Nodal, and bone morphogenetic proteins (BMP). Most cells have TGF-β receptors, and at least one protein isoform can be found in all human tissues. TGF-βs participate in fibrosis, angiogenesis, apoptosis, inflammation, and autoimmunity. Dysregulation of TGF-β expression has been observed in conditions such as pulmonary fibrosis, cirrhosis, Crohn’s disease, and cardiomyopathy [58,77,92,93]. TGF-β promotes tumor progression, metastasis, and chemoresistance in later-stage cancer by inducing EMT [92,93,94,95]. However, paradoxically, in normal and early-stage cancer cells, it represses cell proliferation [77,96]. Chen et al. investigated the role of TGF-β signaling and its interplay with miRNAs in metastatic breast cancer. TGF-β up-regulation increases ZEB1/2, which induces EMT and metastatic capacity. The authors also pointed out the role of TGF-β expression in the tumor microenvironment (TME). Mesenchymal stem cells (MSCs) present in the TME frequently exhibit EMT. MSCs with TGF-driven miR-494 up-regulation facilitate metastasis of BC. On the other hand, hypoxia-inducible factors. The phenomenon of TGF-β overexpression was also observed in NSCLC and was associated with a boosted miR-191 expression that increases TGF-β and promotes cell migration [77,96,97,98]. The study by Huang et al. showed that TGF-β1-activated cancer-associated fibroblasts (CAFs) in breast cancer promote tumor growth and pulmonary metastasis [99]. Furthermore, many tumor cells express cyclooxygenase-2 (COX-2) when exposed to TGF-β1, which correlates with the formation of new vessels within the tumor [100]. Moreover, TGF-β is related to other metastasis-involved targets, such as kinases (mitogen-activated protein kinases, extracellular signal-regulated kinases, c-Jun N-terminal kinases) and receptors (epidermal growth factor receptor-2) [77]. For instance, in breast cancer, the enhancer of zeste homolog 2 (EZH2) intensifies FAK/TGF-β signaling, resulting in bone metastasis [101]. TGF-β also promotes the zinc finger protein SNAI (SNAIL) 1/2 and ZEB1/2 activation [95]. VEGF overexpression in prostate cancer results in an aggressive and invasive phenotype, and TGF-β boosts its expression [102,103,104]. Multiple studies have also investigated the role of TGF-β in the progression of various cancers, including cervix, gastric, kidney, lung, liver, or pancreatic cancer. Results show that, in general, TGF-β is linked to poor prognosis and cancer progression [102]. Nevertheless, there are recent reports on the possible development of TGF-β-targeted agents in antitumor treatment, despite the bipolar nature of TGF-β. Galunisertib, a TGF-β inhibitor, suppressed migration in glioblastoma and the development of pulmonary metastasis [95,105]. It also effectively inhibited the progression of cholangiocarcinoma and induced radiosensitivity in neck squamous cell carcinoma [106]. Additionally, in a melanoma model, it showed anti-metastatic and anti-proliferative activity either in monotherapy or combined therapy with cytotoxic T cell antigen 4(CTLA4) [95,105].

The TGF-β family is important for placental development. TGF-β1 can act as an invasion inhibitor or promoter, and it promotes first-trimester cytotrophoblast differentiation. TGF-β1, TGF-β2, and TGF-β3 suppress EVT invasion, while Activin A, BMP2, and BMP4 boost it. TGF-β1 inhibits trophoblast invasion by regulating the Twist family basic helix-loop-helix transcription factor (TWIST) and SNAIL [107,108,109].What is crucial for this review is a study by Shirakawa et al. on EMT-factor levels in adherent placentas, which revealed elevated expression of TGF-β in the local decidua of the invasive part of the adherent placenta [110]. El-Hussieny et al. investigated villous and EVT cells of normal placentas and PA and reported a significant reduction in TGF-β1 expression. The authors suggest that it is one of the main factors leading to excessive trophoblast invasion [58].

Murrieta-Coxca et al. also reached similar conclusions in their research [109]. However, the results of the study by Khamoushi et al. show increased TGF-β1 in investigated PA myocytes. Nevertheless, the authors support the theory of the crucial role of TGF-β1 in PA pathogenesis [107].

To our knowledge, there are only a few studies that have examined TGF-β1 expression in PA using immunostaining [58,111]. Incebiyik et al. hypothesized that in placenta percreta, the expression of TGF-β will be lowered compared to normal placentas; however, the study result did not show a significant difference [57].

### 2.8. β-Catenin

β-Catenin is an integral component of the Wnt signaling pathway, and it has a significant role in the regulation of cell proliferation, differentiation, and apoptosis. The Wnt signaling pathway is a cellular communication mechanism that has undergone evolutionary conservation and is essential for development, tissue homeostasis, and, when dysregulated, human illness [112,113,114,115].

The human endometrium specifically secretes β-Catenin, and the entire process is estrogen and progesterone dependent. In addition, β-Catenin is involved in the development and differentiation of the endometrium, alongside embryo implantation [58,113]. El-Hussieny et al. observed that immunoexpression of β-Catenin was significantly decreased in placenta accreta (PA) in comparison to control tissue [58]. An immunohistochemistry examination by Han et al. also revealed that the PA group’s placental tissue had lower levels of β-Catenin expression than the control group did [113].

Duzyj et al. noted that lowered β-Catenin immunostaining could suggest increased proteolysis of the trophoblastic cell membrane [46]. These observations imply that decreased expression of β-Catenin could be related to the loss of cell adhesion and possible trophoblast migration [46,58,113].

An imbalance in the structural and signaling properties of β-catenin often results in disease and deregulated growth connected to cancer and metastasis [114,116,117,118,119]. Zhang et al. reported that the Wnt/β-catenin pathway is involved in thyroid cancer progression [117]. Wnt/β-catenin signaling has been linked in some studies to bladder cancer progression, stage, invasiveness, and poor prognosis. Furthermore, it has been proposed that Wnt/β-catenin signaling is essential for the regulation and upkeep of urothelial cancer stem cells [119,120,121,122,123]. The Wnt/β-catenin network plays an important role in hematopoiesis, which is why acute myeloid leukemia (AML), chronic myeloid leukemia, chronic lymphoid leukemia, multiple myeloma, and acute lymphoblastic leukemia (ALL) are among the hematological malignancies associated with abnormal Wnt/β-catenin signaling, either through mutations or ectopic activation [115].

### 2.9. CFL-1

CFL-1 belongs to the actin depolymerizing factor/cofilin family and is the most abundant and ubiquitous member of the family in all studied non-muscle cells and in embryonic muscle cells [124,125]. CFL-1 is crucial for cell survival, as its primary function is to bind and depolymerize filamentous F-actin and prevent the polymerization of monomeric G-actin into filamentous F-actin, resulting in dynamic rearrangement of the actin cytoskeleton [125,126,127].

According to Sparrow et al., CFL-1 is necessary for dynamic changes in the cytoskeleton needed for axon engagement and is essential for Schwann cell myelination, processes involved in nerve growth and repair [128]. Evidence for the involvement of CFL-1 and the Actin-Related Protein 2/3 complex (Arp2/3-complex) in the regulation of axonal growth cones has been recently reviewed by Dumpich et al. [129]. CFL-1 can also participate in the regulation of cell proliferation in response to physical movement or external factors [130].

The actin cytoskeleton plays a major role in the formation of many essential cellular structures such as microvilli and cellular junctions, which transform early pregnancy in mammals. CFL-1 plays an important role in decidualization (the process of preparing the uterus for implantation) and blastocyst implantation [125,127,130]. In a study performed by Ali et al., it was observed that placenta accreta, a medical condition characterized by abnormal attachment of the placenta to the uterine wall, expresses higher levels of CFL-1 in comparison to normal placentation. Moreover, they observed that only CFL-1 had significantly increased levels in focal accreta [127].

Tumor cell invasion and metastasis significantly depend on the remodeling of the cytoskeleton. It is not surprising that CFL-1 pathway abnormalities have been correlated with various types of tumors [131,132,133,134]. Levels of CFL-1 depend on the cancer cell, tumor type, and extent of proliferation and migration. It has been observed that altered CFL-1 expression is present in many tumor samples, such as breast cancer, thyroid cancer, renal cell carcinoma, OC, and oral squamous cell carcinoma [126,131,132,133,134,135]. Additionally, CFL-1 expression is increased in the invasive subpopulation of tumor cells in mammary tumors [135,136]. Furthermore, it has been noted that the activity status of CFL-1 is related to invasion and metastasis [127,132,134,137].

### 2.10. IL-35

IL-35 is a newly identified cytokine in the IL-12 family. IL-35 plays a crucial role in the suppressive function of regulatory T cells (Tregs), but recent findings discovered the trophoblast cells as a constitutive producer of IL-35. IL-35 demonstrates potent immunosuppressive effects [107,138,139,140,141].

IL-35 seems to be a critical player in the maintenance of normal pregnancy, and dysregulation of this cytokine is reported in association with several pregnancy-related complications [107,142].

The serum levels of IL-35 may contribute to the pathogenesis of placenta accreta and could be considered to be a potential target in clinical and diagnostic approaches. In the study by Khamoushi et al., the mean level of IL-35 was significantly increased in patients with placenta accreta compared to healthy pregnant women. It was the first report on IL-35 in patients with placenta accreta [107].

IL-35 is associated with the onset and prognosis of various cancers. It is related to immunosuppressive capacity, by which this interleukin protects cancer cells against apoptosis, enhances angiogenesis, and facilitates cancer progression [138,139,140,143,144,145]. It has been demonstrated that IL-35 is crucial for the development of both benign and malignant tumors, such as prostate cancer (PCA), NSCLC, pancreatic ductal adenocarcinoma (PDAC), advanced breast cancer, and hepatocellular carcinoma (HCC) [138,139,140,146,147]. Studies have shown that IL-35 expression is elevated in NSCLC, and it is highly related to the progression and prognosis of NSCLC [143,147]. Recently, Pylayeva-Gupta et al. reported that IL-35 produced by B-cells has a pro-tumorigenic role in pancreatic cancer through a mechanism involving the IL-35-mediated stimulation of tumor cell proliferation [148].

## 3. Discussion

Both invasive placentation and cancer metastasis involve the migration and invasion of cells into surrounding tissues, and the establishment of new blood vessels to support their growth. In both cases, these processes require a complex interplay of molecular signals that regulate cell behavior, such as growth factors, cytokines, and extracellular matrix components.

For example, in both cases, the hypoxic microenvironment plays an important role in promoting the expression of angiogenic factors such as VEGF and PlGF, which stimulate the growth of new blood vessels. Moreover, in both cases, the expression of matrix metalloproteinases (MMPs) is up-regulated, facilitating the degradation of extracellular matrix components and promoting cell migration and invasion.

Another similarity is the involvement of the immune cells in the regulation of these processes. In the case of invasive placentation, the immune cells such as uterine natural killer cells (uNK) and macrophages are recruited to the site of implantation and play a significant role in remodeling the uterine vasculature and promoting trophoblast invasion. Similarly, in cancer metastasis, the immune cells can promote or inhibit tumor growth and invasion and thus can be a potential target for therapeutic intervention.

In both cases, there is also heterogeneity in the cell populations involved, and not all cells within the placenta or tumor are equally invasive or metastatic. This heterogeneity is thought to arise from genetic and epigenetic differences between cells, as well as from differences in the microenvironment and the signaling cues received by the cells.

Overall, while invasive placentation and cancer metastasis differ in their physiological contexts and evolutionary origins, they share many molecular and cellular mechanisms that underline their ability to invade and proliferate in surrounding tissues.

It is also important that implantation/placentation and cancer progression are species-specific processes, which adds to their complexity and makes studying them very difficult. Some of the data received from animal models should not be uncritically translated into humans. For instance, the mechanisms of trophoblast invasion, and spiral artery remodeling in humans and chimpanzees differ from those in gibbons and baboons [93]. On the other hand, in our review, we have tried to show that at the molecular level, many similarities exist between these two biological phenomena.

The process of EMT and consecutive mesenchymal-to-endothelial transition (MET) is at the center of new vessel formation and arterial remodeling during implantation and placentation. One of the key characteristics of EMT is the gain of migratory behavior of the cells [149]. It is well recognized that the gain of such a migratory phenotype is a prerequisite for the metastatic potential of tumor cells.

As mentioned earlier, VEGF is secreted by various types of human cells and its different forms interact with appropriate types of VEGF receptors (VEGFRs). It was shown that in the process of angiogenesis, VEGFR-2 has a central role, since its expression is largely restricted to endothelial cells and their precursors [150]. The process of angiogenesis is of paramount importance for supplying nutrients and oxygen to the tumor. As a zone of 2 mm in diameter or thickness represents maximal tumor size beyond which its cells start to die because of hypoxia, it is well recognized that with this size, tumor angiogenesis must occur. This process, sometimes referred to as neovascularization, has an additional effect on tumor growth and the clinical course of the disease: it gives access to the vasculature without which the tumor cells wouldn’t be able to spread to distant sites [151].

On the other hand, the formation of new vessels and invasion within the uterine wall is necessary for normal pregnancy. It has been postulated that the same growth factors and molecules are involved in the physiological processes of implantation and placentation as in carcinogenesis. Disruption of this homeostasis at the early stages of pregnancy may lead to numerous complications, such as PAS [58]. Extensive neovascularization is evident in the majority of PAS cases, suggesting a proangiogenic phenotype of syncytiotrophoblastic cells [16].

Invasion and metastasis are biologic hallmarks of malignancy. At the same time, deep myometrial invasion or even perforation through the full thickness of the uterine wall isa typical picture for placenta increta and placenta percreta, the most serious forms of PAS.

Both pathological conditions need the ability of cells to overcome the local mechanisms of immunological defense, activate invasion, and induce angiogenesis.

In the case of invasive tumors, it is believed that certain cell subclones gain the combination of gene products to complete all the steps required for metastasis. Such abnormalities give the tumor a general predisposition for metastasis, which has been called a “metastasis signature”.

We have hypothesized that this molecular signature may be involved not only in tumor progression but also in PAS (Table 1).

The biomolecules presented above are involved in different pathophysiological pathways in carcinogenesis and PAS, which we tried to show in Figure 1.

## 4. Conclusions

Given the striking similarities between the pathogenesis of PAS and metastasis, we analyzed molecules that have been widely reported in both oncology and obstetric research as being involved in the microenvironment of both tumors and placental cells.

We hope that our review, on the one hand, will contribute to a better understanding of both these conditions, and, on the other hand, will make it clear how extremely serious an obstetric problem PAS is.

Due to similar results in both pathologies, most of the presented biomolecules deserve special attention. These particles include PlGF, CDH1, LAMC2, ZEB proteins, αVβ3 integrin, CFL-1, and Il-35. Nevertheless, further research is required.

## Figures and Tables

**Figure 1 cancers-15-02618-f001:**
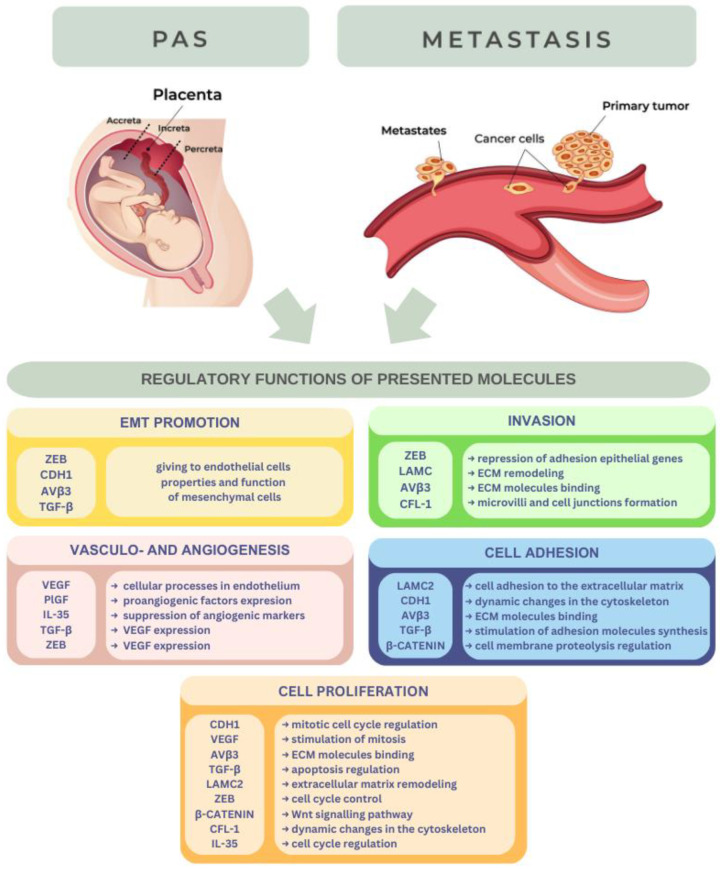
Biomolecules involved both in metastasis and PAS.

**Table 1 cancers-15-02618-t001:** Biomolecules involved both in metastasis and PAS.

Biomolecule	Metastasis		PAS		References
PlGF	↑	increased level	↑	increased level	[14,19,20,21,22,23,24,25,26,27,28]
VEGF	↑	overexpression	↓	decreased level	[16,19,33,34,38]
CDH1	↓	decreased level	↓	decreased level	[43,45,46,50,51,57,58]
sFLT-1	↓	decreased level	↑	increased level	[16,19,27,28,31,38]
LAMC2	↑	overexpression	↑	overexpression	[59,63,68]
ZEB1	↑	ZEB1 promotes metastasis in various cancers	↑	overexpression	[77,78,79,83,84]
ZEB2	↑	overexpression	↑	ZEB 2 promotes EMT thus cytotrophoblast differentiation into invasive EVT	[72,75,82]
αVβ3 integrin	↑	overexpression	↑	overexpression	[6,85,87,88,89,90,91]
TGF-β	≠	TGF-β enhances metastasis in later-stage cancer, represses in early-stage cancer	≠	Increased/decreased	[58,77,96,97,98,99,102,103,104,107,109]
β-Catenin	↑	overexpression	↓	decreased expression	[46,58,62,113,116,121]
CFL-1	↑	overexpression	↑	increased levels	[127,131,135,136]
IL-35	↑	overexpression	↑	increased levels	[107,138,139,140,146,147]

CDH1—E-cadherin; CFL-1—cofilin-1; IL-35—interleukin-35; EMT—epithelial-to-mesenchymal transition; EVT—extravillous trophoblast; LAMC2—laminin γ2; PAS—placenta accreta spectrum; PlGF—placental growth factor; TGF-β—transforming growth factor β; VEGF—vascular endothelial growth factor; ZEB—zinc finger E-box-binding homeobox proteins; ↑—up-regulation; ↓—down-regulation; ≠—ambiguous.

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
