# Peer review of "Biomolecules Involved in Both Metastasis and Placenta Accreta Spectrum—Does the Common Pathophysiological Pathway Exist?"

_cancers, 2023, doi:10.3390/cancers15092618_

Round 1
Reviewer 1 Report
Dear authors and editor,
The manuscript titled “Biomolecules involved both in metastasis and placenta accreta spectrum - does the common pathophysiological pathway exist?” describes a number of molecular factors involved in the intricate migration and invasion of cells into the surrounding tissues. This process called EMT (epithelial-to-mesenchymal transition) occurs physiologically(implantation of blastocyst) as well as in pathological situations ( metastasis of caner). The authors rightly noted the similarities of both processes and analyse selected factors involved in this process ( placental growth factor (PlGF), vascular endothelial growth factor (VEGF), E-cadherin (CDH1), laminin γ2 (LAMC2), the zinc finger E-box-binding homeobox (ZEB) proteins, αVβ3 integrin, transforming growth factor β (TGF-β), β-catenin, cofilin-1 (CFL-1), and interleukin-35 (IL-35)). Table 1 and Figure 1 present the results of the analysis in a very clear way.
There are many similar studies in the literature. In the cited publications, I have not found any previous publications by authors in this field. Looking at the list of authors, I trust that one of the goals of the presented manuscript is thorough preparation for the implementation of the original scientific project. If that was the concept then the manuscript fulfils its function.
The paper is generally well-organised, the language is correct and the content is understandable. Minor language errors to be corrected during possible preparation for publication. Literature properly selected and up to date. Importantly, over 90% of citations come from the last 10 years.
In conclusion, I support the publication of the manuscript.
Thank you for your choice me as a reviewer.
Reviewer 2 Report
This is an excellent, comprehensive and timely review, comparing and contrasting PAS with tumor metastasis.
Reviewer 3 Report
The authors well-described the parallel between cancer metastasis and PAS, an obstetric disorder. The review is very interesting and well-written.
No major issues are arosen, but I would suggest to better connect Figure1 to the text. In addition:
-Include a Gifure with biological processes interconnection based on the target biomolecules
- Eliminate Sentence: line 595-597
Reviewer 4 Report
The review article by Rekowska et al, describes the biomolecules that are involved in both placenta accreta spectrum and in cancer metastasis. The review is very well explained and covers the important molecules that play major role in both the process. I have no further corrections to add to the manuscript.
